# The Therapeutic Potential of a Strategy to Prevent Acute Myeloid Leukemia Stem Cell Reprogramming in Older Patients

**DOI:** 10.3390/ijms241512037

**Published:** 2023-07-27

**Authors:** Moon Nyeo Park

**Affiliations:** Department of Pathology, College of Korean Medicine, Kyung Hee University, Hoegidong Dongdaemungu, Seoul 05253, Republic of Korea; mnpark@khu.ac.kr

**Keywords:** AML, LSCs, herbal medicine, natural product, miRNA, epigenetic modifications

## Abstract

Acute myeloid leukemia (AML) is the most common and incurable leukemia subtype. Despite extensive research into the disease’s intricate molecular mechanisms, effective treatments or expanded diagnostic or prognostic markers for AML have not yet been identified. The morphological, immunophenotypic, cytogenetic, biomolecular, and clinical characteristics of AML patients are extensive and complex. Leukemia stem cells (LSCs) consist of hematopoietic stem cells (HSCs) and cancer cells transformed by a complex, finely-tuned interaction that causes the complexity of AML. Microenvironmental regulation of LSCs dormancy and the diagnostic and therapeutic implications for identifying and targeting LSCs due to their significance in the pathogenesis of AML are discussed in this review. It is essential to perceive the relationship between the niche for LSCs and HSCs, which together cause the progression of AML. Notably, methylation is a well-known epigenetic change that is significant in AML, and our data also reveal that microRNAs are a unique factor for LSCs. Multiple-targeted approaches to reduce the risk of epigenetic factors, such as the administration of natural compounds for the elimination of local LSCs, may prevent potentially fatal relapses. Furthermore, the survival analysis of overlapping genes revealed that specific targets had significant effects on the survival and prognosis of patients. We predict that the multiple-targeted effects of herbal products on epigenetic modification are governed by different mechanisms in AML and could prevent potentially fatal relapses. Thus, these strategies can facilitate the incorporation of herbal medicine and natural compounds into the advanced drug discovery and development processes achievable with Network Pharmacology research.

## 1. Introduction

AML has numerous morphological, immunophenotypic, cytogenetic, biomolecular, and clinical characteristics resulting from genomic and epigenetic alterations [1,2,3]. Despite the progress that has been made in treating AML, recurrence continues to be the most challenging problem. Even though 10–40% of AML patients younger than 60 years old are generally resistant to AML induction therapy, the rate for patients older than 60 years old is much greater, ranging from 40 to 60% [4]. The basic rationale for this approach is that AML has been recognized as the etiological agent that first provokes the development of the cancer stem cell. Thus, extensive research on the genetics of AML has led to the sustained use of stem cell therapy for the disease [5]. Furthermore, LSCs are derived from leukemia-initiating cells at various phases within primitive multipotent cells, which may cause the leukemic clone to relapse because leukemic cells are diverse [6,7,8]. Allogeneic stem cell transplantation (AlloSCT) is the only long-term cure for myeloid malignancies such as AML, chronic myeloid leukemia (CML), myelodysplastic syndromes (MDS), chronic myelomonocytic leukemia (CMML), and juvenile myelomonocytic leukemia (JMML), especially in high-risk individuals [9]. Age and comorbidities, such as those calculated by the Hematopoietic Cell Transplantation-specific Comorbidity, affect transplant potential [10]. After a few cycles of induction chemotherapy (IC), 30% of patients with recently diagnosed AML do not experience morphological complete remission (CR) [11]. The fact that no one recognizes how to deal with AML that does not conform with IC makes this quite unsatisfactory [12]. There is no single approach that is commonly regarded as the gold standard for treating AML in the elderly. A few common ways to treat cancer are best supportive care (BSC) alone, standard IC, and low-dose Ara-C (LDAC). The National Comprehensive Cancer Network (NCCN) guidelines indicate that individuals with AML who are younger than 60 and have better prognostic factors should achieve IC. However, many elderly AML patients do not comply with these requirements [13,14]. The main problem with developing targeted inhibitors for the therapeutic management of AML is that it differs so greatly between individuals. As a result, tailored therapy is the most effective way to efficiently control this disease [15]. Personalized treatment focuses on the main categories that each patient’s leukemia cells adhere to. Without functional research, genomic approaches are unable to explain how the disease pattern contributes to the disease; they can only identify the genes with significant defects [16]. AML’s classification, identification, prognosis, and treatment are determined by the gene mutation’s cause and cytogenetic profiling’s understanding of its association with LSCs.

## 2. Characterization of Genetic Mutation in AML

Using next-generation sequencing, numerous recurrent somatic mutations in over 90% of AML patients were identified [17,18]. Several proteins, including Nucleophosmin (NPM1), FMS-like tyrosine kinase 3 (FLT3), Runt-related transcription factor (RUNX1), DNA methyltransferase 3A (DNMT3A), Isocitrate dehydrogenase (IDH), Ten-eleven translocation 2 (TET2), and CCAAT/enhancer-binding protein-alpha (CEBPA), have been identified [17,18,19]. AML patients had frameshift and cytoplasmic protein elongation in the NPM1 gene exon 12 on chromosome 5q35 [20]. In AML patients, deletion/insertion mutations in NPM1 exon 12 (NPM1-DIM) change the C-terminal amino acids and impair the protein’s nucleocytoplasmic shuttling activity, causing disease pathogenesis [21]. A total of 30% of de novo AML patients have NPM1 mutations [22]. Therapy-targetable molecular subgroups determine prognosis. Diploid karyotype patients sometimes have unusual mutation interactions. A total of 50% of diploid karyotype individuals with an NPM1 mutation develop an FLT3 mutation, most commonly FLT3 internal tandem duplication (FLT3-ITD) [23,24]. Thus, the FLT3 allele ratio dictates the outcome, which historically has been poor [25]. From 25 to 45% of AML patients have the FLT3 mutation [22]. The FLT3 gene mutation activates FLT3, a tyrosine kinase receptor, in AML blasts. This stimulates leukemogenesis and controls leukemic blasts [26]. A total of 70% of exon 14 FLT3-ITD in-frame mutations were found [27]. In AML, not otherwise specified subcategories have no prognostic value when classified by NPM1 mutation and CEBPA biallelic mutation status; therefore, acute erythroid leukemia, erythroid/myeloid type has been removed from the category [28,29]. Due to chromosomal translocations and point mutations, the AML1/RUNX1 gene (RUNX1) is one of the most frequently uncontrolled genes in leukemia. It has 10 exons (1–6, 7A–7C) [30,31]. The exon 4 (c.497_498insGA) frameshift RUNX1 mutation was found in all patients’ DNA [32]. The ETO/MTG 8/RUNX1T1 gene on 8q22 frequently translocates with the AML-ETO gene to form the fusion protein AML-ETO AML [33]. Most AML patients have numerous DNMT3A mutations at codon R882. Mutations, including R882, have been found on exons 8–23 [34,35]. Only the key element DNMT will be addressed here. Three isoforms of IDH convert isocitrate to -KG. IDH1, IDH2, and IDH3 live in the cytoplasm, mitochondria, and mitochondrial matrix, respectively [36]. Gliomas [37,38], cholangiocarcinoma, chondrosarcoma, and chondromas [39] carry IDH1 and IDH2 mutations [39]. Missense variants that substitute arginine residues at codon 132 in exon 4 of the IDH1 gene and codons 140 or 172 in the IDH2 gene generate IDH1/2 mutations [40]. AML prognosis is affected by a germline-synonymous SNP in exon 4 of the IDH1 gene, codon 105 [41,42]. IDH2 mutations are beneficial in all non-M3 or cytogenetically normal AML, but IDH1 mutations decrease overall mortality and event-free survival, especially in patients with normal cytogenetics [43,44]. TET2 converts 5-methylcytosine to 5-hydroxymethylcytosine (5hmC), 5-formylcytosine, and 5-carboxylcytosine for hematopoiesis and HSCs differentiation. TET2 exon 3 mutations lower 5hmC, enhance HSCs self-renewal, and expand myeloid lineage cells, causing AML [45]. The CEBPA gene has a single exon with a GC-rich (>70%) coding sequence matched to chromosomal band 19q13.1 [46]. AML with biallelic CEBPA mutation had a new frameshift mutation, c.1590delC [35]. Although treatment has greatly improved the prognosis for young AML patients, the majority of new cases are elderly people whose prognosis is quite dismal. The 5-year relative survival rate is 30–50%, even with current therapies [47]. There is no easy solution, and choosing the best method is difficult. Characterizing LSCs may assist in identifying patients who will benefit from innovative targeted therapy [48]. In order to better comprehend the therapeutic benefits of clinical trials for disease treatment, the purpose of this review is to identify and characterize relapse and resistance due to the dysregulation of pathways and environments with LSCs in AML (Figure 1).

## 3. Significance of the Phenomenon Caused by HSCs and AML’s Initial Interaction

AML was one of the earliest cancers in which a stem cell was found. AML is characterized by the presence of numerous blast cells, including myeloblasts, monoblasts, and megakaryoblasts, in the bone marrow and/or peripheral circulation. Due to the rapid progression of AML, patients should undergo rapid cytogenetics and molecular analysis to determine optimal risks and treatments [49]. Furthermore, relapse is the greatest barrier to treating AML, as there are no effective treatments, and 90% of people who experience it die. It is now well acknowledged that a persistent subset of AML cells called leukemia-initiating cells or LSCs are responsible for relapse [50]. AML in the initiation of the disease by stromal cells and immune cells through paracrine and autocrine signals for long-term hematopoietic regulation of [51] stem cell-released factors and cytokines that defend AML cells against chemotherapeutic agents and promote drug resistance [52,53,54]. Although LSCs are a very rare subgroup of AML cells, they have the unique ability to establish and sustain a cellular hierarchy, and their persistence throughout remission has been associated with a poor clinical prognosis [55,56,57]. It was interestingly revealed that circulating cancer cells target the “niche” in the bone marrow that is home to hematopoietic stem cells (HSCs) and, more importantly, that they compete with HSCs for occupancy of that niche [58]. HSCs are known to associate with at least two distinct niches in the bone marrow, the endosteal niche (which contains osteoblasts) and the vascular niche (which contains endothelial cells) [59]. Regulation of LSCs and HSCs involves contact-dependent (surface receptors and ligands) and contact-independent (cytokines, chemokines, growth factors, exosomes) interactions with the rest of the bone marrow microenvironment [60,61]. HSCs adapt blood production to the organism’s requirements using cues from the bone marrow (BM) niche. Blood cancers alter the BM niche in a way that directly affects the LSCs that initiate the disease [62]. Cancer cells and leukocytes, including lymphocytes, tumor-associated macrophages (TAM), cancer-associated fibroblasts, endothelial cells, and pericytes, form a diverse population within the tumor microenvironment, contributing to a dynamic and complex ecology [63]. Atypical phenotypes caused by genetic and epigenetic changes to cells’ metabolism include adaptive modifications to signal transduction and transcriptional regulation, which supply the energy and biomass needed for cell proliferation [64]. When cancer cells first connect to the endothelium, N-cadherin plays a critical part in mediating cellular contact. This heterotypic contact is the initial event in the chain of events leading to the metastatic spread of cancer cells through transendothelial migration [65]. Here, it was observed that tumor stiffness modifies the CCN1/catenin/N-cadherin pathway, which, by making it easier for cancer cells to adhere to blood vessels, contributes to the metastatic cascade [66]. Cysteine-rich angiogenic inducer 61 (CYR61) is a matricellular protein expressed by endothelial cells in stiff surroundings that activates β-catenin to increase N-cadherin expression. This allows cancer cells to enter the bloodstream and metastasize through stable endothelium interaction [66,67]. Differentiating and targeting LSCs from HSCs has been made possible by the identification of cell membrane surface markers, such as CD33 [68], CD123 [69], CLL-1 [70], CD44 [71], CD47 [72], and CD96 [73], which are expressed preferentially on LSCs compared to HSCs in AML [74]. The percentage of HSCs separated from a leukemia patient that carries at least the first mutation is what we refer to as the “preleukemic burden” in order to measure this heterogeneity. A total of 39 individuals with AML were studied to determine the frequency of known leukemogenic driver mutations in HSCs, T cells, and blasts [75]. It was predicted that TIM3 (CD366), a member of the T cell immunoglobulin mucin (TIM) family, is a novel, important, and differentiating membrane surface marker for LSCs [76,77,78,79,80], a negative regulator of T helper1 (Th1) cells immunity [81], and a prognostic factor in patients with AML [74]. High levels of TIM3 on T cells in the peripheral circulation mediate exhaustion/dysfunction of T cells in AML patients. This may be a crucial immune surveillance strategy for leukemia [82]. LSCs have many characteristics in common with their normally functioning counterparts, such as quiescence, which is characterized by low metabolic activity [83], high expression of anti-apoptotic proteins like Bcl-2 [84], and resistance to cellular stress and cell death caused by chemotherapeutics [50]. As previously discussed, cancer cells can enter a quiescent state and wait until their fitness and environmental conditions are conducive to growth in order to keep the disease undetected for a long time [67,85]. Stromal cells surround the sinusoidal blood vessels and are essential to HSCs; it has been identified that CD271 and CD146 are markers for these cells [86,87,88]. By identifying these mutational patterns, it may be possible to cure or avoid these mutations using therapies such as HSCT, in addition to estimating patient outcomes based on AML subtype and prognostic variables [89]. Without cell division, bloodstream survival, or homecoming, LSCs increased. The self-renewal-related genes KLF4, Bmi-1, and Nanog are also expressed by LSCs, which have the ability to form sphere-like structures in vitro and weak tumors in vivo [90]. Additionally, tumor-promoting stem cell signaling pathways such as Wnt, Nanog, and Oct4 were activated by extracellular matrix (ECM) components in LSCs, resulting in the spread of metastasis [91,92]. It is becoming obvious that the microenvironment interacts with leukemic blasts, develops malignant cells that support LSCs’ immune escape mechanisms, and inhibits effector cell immunocompetence, such as T and natural killer cells [93]. The perivascular niche is a tumor-promoting environment that is made up of a variety of micro-vessels. It is responsible for regulating the dormancy of cancer cells that have spread from various primary tumors to the bone marrow, the lungs, and the brain [94,95,96,97]. The abundant availability of oxygen, nutrients, and paracrine substances found in perivascular niches makes them an ideal habitat for the growth of CSCs [98,99]. Dormant cells that have been adapted through genetic and epigenetic editing [85] may contribute to disease evolution in regards to phenotype and function by promoting TME remodeling, making it “fertile soil” for propagation [63]. Only cells grown in a soft matrix express CSC markers [100], suggesting that a flexible microenvironment may enhance cancer cell stemness, an attractive hypothesis that needs in vivo validation [66]. Recent research has identified that CSCs can be relatively abundant (at least in some tumors), have the ability to switch between dormant and proliferating states, and are characterized by a high degree of heterogeneity and plasticity over space (that is, in different regions of the tumor) and time [101,102]. In general, this complex molecular cross-talk between the LSCs and its gaps can be controlled by acquiring a comprehension of the interaction that maintains the LSCs in a quiescent state. Epigenetic changes result in recurrence during the cytogenic compensated process. Although epigenetic modifications such as DNA methylation are crucial, it is believed that identifying factors other than well-known mutations could bring the possibility of treatment closer (Figure 2).

## 4. Epigenetic Modification of the LSCs Microenvironment

Tumor somatic cells can become chemotherapy-resistant through genetic changes. Cancer cells may develop treatment resistance through multidrug resistance, apoptotic suppression, epigenetic metabolism, DNA repair, and gene amplification [103]. Cancer stem cells and normal stem cells share stem cell markers [104,105]; hence, genetic abnormalities that cause cancer, such as oncogene acquisition or DNA mutations, should determine differentiation [106,107]. Cancer cells heavily exploit epigenetic markers like DNA methylation and microRNA to modulate gene expression. Despite decades of therapeutic success, progressive and repeating genes and non-coding expression patterns remain mysterious. Wnt/β-catenin, Hedgehog, and Notch signaling regulate normal stem cells and promote or inhibit cancer cell proliferation [108,109,110,111]. Mutations that activate conventional stem cell pathways can help cancer stem cells develop. Hematopoietic and brain stem cells self-renew via Wnt and Bmi1 pathways [112]. p16 INK4a, p19ARF, RB, and p53 govern cell cycle entry and exit. For normal cells to become cancerous, p16 INK4a, p19ARF, RB, and p53 must be inactive [113]. Many tumors, particularly hemopoietic malignancies, remove or inactivate the phosphatase and tensin homolog (PTEN) [114], including hemopoietic malignancies [115,116,117]. Mutations in these genes can alter gene expression, turning healthy stem cells into cancer stem cells that drive tumor growth. RNA splicing, DNA methylation, chromatin remodeling, transcription factors, activated signaling, cohesion complexes, tumor suppressors, and nuclear phospholipids are connected with many function-based subgroups [118,119,120]. Hematopoiesis requires DNA methylation [121,122]. Hematological malignancies cause multiple epigenetic changes, including hypo- or hyper-methylation of CpG islands in promoter regions [123], a well-known epigenetic regulation in carcinogenesis [124]. CpG islands show that differential methylation influences miRNA gene expression. Several candidates had malignancy-associated hypermethylation and downregulation [125]. Hub genes and possible signaling pathways involved in the recurrence of AML were studied. Two gene and non-coding RNA expression profiles were taken from the Gene Expression Omnibus (GEO) database. TNF, IL6, TLR4, VEGFA, PTPRC, TLR7, TLR1, CD44, CASP1, and CD68 were the hub genes [126]. DMRs at microRNA loci accidentally dysregulate many targets, including cancer-related master genes [127]. Epigenomic and transcriptome data identified DMRs, CIMPs, and cancer-associated miRNAs. DNA methylation is dynamic and reversible, which may affect cancer treatment [128]. HIF-1, CTCF motif, MYC, LAMB3, RUNX2, and Oct4 are susceptible to CpG-methylated recognition site sequences, which may affect miRNAs via RNA polymerase II [129,130]. DNA methylation, processing, cancer stem cell features, and microRNA promoters affect microRNA expression, according to recent studies. A recent study found that the miR-200 family, miR-9 family, miR-200b-200a-429 cluster, and their target mRNAs regulate CSC DNA methylation [131,132,133,134,135]. All genomes had mRNA and miRNA expression data. Fusion protein and DNA methylation patterns strongly correlated with AML sample expression signature and differentiation status [136,137]. Global DNA demethylation is necessary for early embryos to become pluripotent and primordial germ cells (PGCs) to lose parental-origin markers [138]. There is accumulating evidence that replication-dependent passive loss of the fifth position of cytosine (5-methylcytosine, 5mC) cannot explain the fast loss of 5mC during these two major waves of epigenetic reprogramming. Enzymatic mechanisms may actively remove or modify cytosine methyl groups [139]. 5mC, a conserved epigenetic alteration, is present in most plant, animal, and fungal models [140]. It is widely known that epigenetic modulators like non-coding RNAs regulate the epithelial-to-mesenchymal transition (EMT) and cancer stem cells (CSCs), which overlap cancer cell invasiveness [141]. We investigated EMT-induced stemness mediators by profiling microRNA expression and DNA methylation at microRNA promoters. MiR-203 overexpression causes differentiation and decreases stem cell characteristics [131,142], and miR-663 also affects ATRA-induced HL-60 cell differentiation. Lentiviral-mediated miR-663 can effectively cure hematological malignancies [143]. Immunological escape or immunosurveillance can cause cancers. This study analyzes how cancer reprogramming of immune and tumor cells affects epigenetic alterations such as DNA methylation and histone protein levels [144] and miRNA. The current work examined epigenetic modifications in AML signaling, miRNA, and DNA methylation to determine whether miRNA may be used to treat AML. The epigenome is influenced by key metabolites dynamically regulating LSCs metabolism (Figure 3).

## 5. Intelligent Application: Let the Fat out and Use It as Fuel

Surprisingly, the majority of LSCs that can be identified functionally are “ROS-low”. BCL-2 inhibition decreases oxidative phosphorylation and eliminates quiescent LSCs in ROS-low LSCs [145]. Despite the fact that it has been presumed, at least in part, that the biology of LSCs is similar to that of normal tissues, the majority of LSCs in patients with AML are not quiescent (G0/G1: 80–90% with 0.5–10% in G0) [146,147]. When CSCs transition from a quiescent stage to a highly proliferating stage, glycolysis appears to be crucial [148]. Thus, it was found that amino acid or fatty acid oxidation (FAO) creates the high-energy hydrogens (NADH+H^+^ and FADH2) that drive oxidative phosphorylation (OXPHOS) in AML, which also requires efficient mitochondria [149,150]. LSCs sustain lipolysis in adipocytes, which generates free fatty acids that enhance β-oxidation in AML cells, thereby enhancing their survival, proliferation, and transformation [151]. Thus, AML bone marrow continuously assimilated glucose, whereas pyruvate, lactate, and glycerol-3-phosphate were inversely related to survival [152]. Here, leukemia increases lipoxygenases arachidonate 15-lipoxygenase (Alox15/15-LO (ALOX15) and ALOX5), which metabolize polyunsaturated fatty acids [107]. Increased mitochondrial mass1 and altered metabolic characteristics, such as a greater reliance on OXPHOS [153,154], and FAO [155], are characteristics of this altered phenotype. When compared to normal tissues, cancer cells have a highly stimulated de novo production of these elongated FA chains, which are structural elements of membranes [156,157]. Moreover, many critical signaling pathways, such as p53 [158], Nuclear factor-κB (NF-κB), and Phosphoinositide 3-kinases (PI3Ks) [159], have been connected to Alox5 function [160]. Furthermore, the identification of IDH1 and IDH2 mutations in cancer further emphasizes the direct relationship between metabolic instability and carcinogenesis. These alterations join loss-of-function variants that target fumarate hydratase (FH) and various succinate dehydrogenase (SDH) subunits [161,162]. The metabolic enzymes FH, SDH, and IDH have been altered by cancer. Scientists believed that abnormal cancer cell proliferation was brought on by metabolic problems. According to this hypothesis, metabolic reprogramming in CSCs can cause cancer without DNA changes in cancer genes [163]. Given that mutations targeting FH and SDH result in similar increases in prolyl hydroxylases (PHDs), Hypoxia-inducible factor-1 alpha (HIF-1 α) is highly intriguing [164,165]. Multiple studies have shown that Nanog and Sox2, two of the most important transcription factors in cancer stem cells, induce cancer cell metastasis in response to hypoxia. Nanog binds directly to the enhancer region of the BNIP3L (BCL2 Interacting Protein 3 Like) promoter during hypoxia, thereby releasing BECLIN from Bcl-2 (B-cell lymphoma 2), increasing its expression, and promoting one of the hypoxia-induced responses of cancer cells [141,166]. AML patients have altered succinate levels and dysregulated TCA cycle activity. Succinate modifies epigenetic activity by inhibiting 2-oxoglutarate (2-OG)-dependent histone and DNA methylation enzymes [167]. Between succinyl-CoA and isocitrate, alpha-ketoglutarate (KG) is a critical intermediary in the tricarboxylic acid (TCA) cycle. As a crucial component of the anaplerotic process, KG controls ATP synthesis and decreases equivalent (NAD+/NADH) production in the TCA cycle, which affects the level of ROS and immune system homeostasis [168]. Fe (II)/α-KG-dependent dioxygenases, which are widely distributed in all living organisms [169,170], are thought to use α-KG as a co-substrate. Only a few of the biological processes in which these enzymes play a key role include the post-translational modification of collagen, fatty acid metabolism, oxygen sensing, DNA and RNA repair, and demethylations associated with epigenetic regulation [171]. According to research, alpha-KG maintains the pluripotency of embryonic stem cells (ESCs) [172]. Moreover, α-KG is a primary source of glutamine and glutamate, which are required for the synthesis of both amino acid and collagen [173]. Glutamine is important for cancer stem cell maintenance because it repairs DNA damage through nucleotide biosynthesis and a redox-mediated mechanism [174]. Based on the above-mentioned, it has been determined that AML influences the corresponding process, FAs metabolism [175,176], and mitochondrial oxidative phosphorylation [177,178,179,180] during the disease by requesting preferential lipid metabolization over glucose or glutamine [175,176]. Understanding LSC- and HSC-quiescence is crucial for selecting future AML treatment targets, as quiescent AML cells are responsible for the majority of relapses [150]. The process of reprogramming cells into cancer stem cells has been observed to significantly exacerbate relapse, drug resistance, and metastasis in a diverse range of patient cancers. The aforementioned analysis offers a heightened comprehension of HSCs and LSCs, thereby empowering researchers to focus specifically on the LSCs subset by considering the distinctive attributes of AML in the context of Figure 4.

## 6. Herbal Treatment Strategies for AML Employing System Biology

Rydapt, an inhibitor of Fms-related receptor tyrosine kinase 3 (FLT3), and venetoclax, an inhibitor of Bcl-2, are two examples of inhibitory biological therapies used to treat AML [181,182]. Nowadays, genetic and epigenetic pharmaceuticals can treat the genetic etiology of disease as opposed to the gene mutation [183]. Given the complexity of AML, we suggest a candidate group that uses herbal medicines, including the aforementioned components, to target many factors at once. Due to their various efficient constituents, herbal remedies have “multitarget and multipath effects”. Numerous studies have demonstrated the synergistic effects that formulations frequently have, which are likely to have an impact on a variety of biological functions and intracellular pathway networks [184,185]. Drug screening is sped up, made more affordable, and improved through computer modeling of biological systems [186]. This facilitates drug discovery through herbal and pharmacological understanding [187,188]. Recent studies have shown that miRNAs have a crucial role in controlling the production of oncogenes or tumor suppressors, which in turn controls biological processes such as apoptosis, proliferation, and differentiation in hematopoietic cells [189,190]. If we can determine which herbal medicine mechanism modulates miRNA and AML-related target factors, we could provide a candidate group for a long-term AML therapeutic alternative. Over 300 prescriptions contain *Leonurus japonicus* Houttuyn (LJH) in traditional Korean medicine. LJH increased reactive oxygen species, and miR-19a-3p was suppressed in direct relation to PTEN, which increased apoptosis [191]. Dragon’s blood, also called *Daemonorops draco* Blume (DD), is a traditional Korean medicine used to relieve pain from wound infections. DD increased ROS generation and miR-216b expression while downregulating c-Jun [192]. Casticin has anti-inflammatory and anti-carcinogenic properties that are used in traditional herbal medicine. MicroRNAs, which control oncogenes and cancer suppressors, are dysregulated in diseases. Through upregulating miR-338-3p, which targets RUNX2 and inhibits the PI3K-Akt signaling pathway, casticin increases AML cell death but reduces cell proliferation in vitro and tumor growth in vivo [193]. Qinghuang (QHP) powder and various mixtures are utilized to treat blood cancers such as AML. QHPs were associated with important targets such as AKT1, MAPK1, MAPK3, PIK3CG, CASP3, CASP9, TNF, TGFB1, MAPK8, and TP53 using systematic docking and molecular docking visualization [194]. PTL-mediated apoptosis is significantly linked to NF-B suppression, p53 activation, and enhanced ROS by targeted therapy targeting LSCs in primary AML cells, bcCML cells, normal bone marrow (BM), and umbilical cord blood (CB) cells from the National Disease Research Interchange (NDRI) [195]. Quercetin activated caspase-8, caspase-9, and caspase-3, cleaved PARP, and depolarized mitochondria via ERK-mediated apoptosis pathway in THP-1, MV4-11, U937, and HL-60 cells [196]. Parts of the fruit of the medicinal tree Melia azedarach have been identified as having anti-cancer properties. 1-Cinnamoyltrichilinin (CT) significantly decreased the viability of AML cells. CT has induced apoptosis in AML cells and activated the p38 pathway [197]. Mitochondria membrane potential (MMP) is interfered with by curine, a natural alkaloid obtained from Chondrodendron platyphyllum. Curine has also been shown to have strong cytotoxic effects on leukemic cell lines [198]. Produced by plants, algae, bacteria, and fungi, carotenoids are organic pigments. Several carotenoids have demonstrated cytotoxic effects on AML cells [199]. Cactus pear fruit indicaxanthin (Ind) inhibits NADPH oxidase-4 (NOX-4) basal activity and overexpression, NF-κB activation, cell redox balance, Ca homeostasis, mitochondrial damage, and apoptosis [200]. Fucoxanthin increases apoptosis cleavage of caspases-3 and -7 and PARP via Bcl-xL [201]. Actinodaphne lancifolia, a plant used in traditional oriental medicine, has a substance called isolancifolide, which was isolated from it. Investigations into apoptotic activity included the release of cytochrome c, Smac/DIABLO, and cleaved caspases-8, -3, and PARP [202]. Phenolics have hydroxyl groups directly linked to aromatic hydrocarbon groups. As the most abundant phytochemical, this is highly investigated [203]. Curcumin strongly phosphorylated ERK, JNK, and c-Jun and Jun B. Caspase activity and ERK/JNK/Jun cascade activation showed that JNK and ERK inhibitors inhibited curcumin’s pro-apoptotic effect on THP-1 cells [204]. Using the peculiar and complex molecular characteristics of leukemia cells, we provide adequate support for herbal medicine, a potential group that may be involved in the selective targeting of LSCs via a number of strategies. However, our limited understanding of the mechanisms of action of many of these natural compounds is one of the primary obstacles preventing the widespread adoption of herbal medicine in contemporary healthcare [205]. Vitality and homeostasis are returned to the body through herbal medicine [206]. Until now, drug research and development has targeted one target—a receptor, ion channel, enzyme, or regulatory protein—to treat a disease or symptom. Conventional drugs are increasingly embracing polypharmacology as a supplement to using a single active ingredient or component to treat a specific target because the human body is a complex, interrelated system of cells, tissues, and organs [207,208]. Systems biology techniques, although still in their earliest stages, may offer a mathematical framework for usefully combining the growing quantity of data and information obtained during cancer diagnosis and help design logical defenses to herbal treatment [209]. The research and application of herbal medicine, which has numerous chemical constituents and molecular targets, is ideal for network pharmacology because it emphasizes multi-channel modulation of signaling pathways [210]. Therefore, it can be observed that herbal medicines and natural substances have exhibited promising characteristics as inhibitors of AML in the field of pharmaceutical research, as indicated in the provided table. Additionally, our investigation revealed the effects of chemoprevention, which have been extensively studied over the past decade. In the following section, clinical trials are conducted to determine how adding additional medications to HMA treatments can improve them (Table 1).

## 7. Clinical Research and AML Treatment

For decades, therapy has been divided into two categories: intensive and non-intensive. In intensive chemotherapy (IC), anthracycline + cytosine arabinoside (araC) backbones are used, followed by consolidation with further chemotherapy or allogeneic stem cell transplantation (SCT) [211,212]. Azacitidine (AZA) and decitabine are recommended by the NCCN for older adults with newly diagnosed AML. There was recent European approval of decitabine for over-65s with newly diagnosed AML (20% BM blasts) who are not candidates for standard IC [213]. Thus, AZA and a BCL-2 inhibitor (venetoclax; VEN) can disrupt the TCA cycle, decrease OXPHOS, and impair energy production [214]. Notably, the efficacy of VEN-based therapy for AML patients was compared to standard chemotherapy, and factors and mechanisms involved in VEN sensitivity and resistance in AML cells were examined to identify response biomarkers and combination therapies that could increase AML cell sensitivity to VEN. VEN works better for NPM1-, IDH1/2-, TET2-, and RUNX1-m. Mutated NPM1 is called NPM1c and localizes in the cytoplasm instead of nucleoli [20,215]. The observed overall response rate (ORR) of 19% for VEN in AML was found to be adequately tolerated [216]. A pivotal Phase Ib clinical investigation was conducted to evaluate the efficacy and safety of administering either HMA (azacitidine or decitabine) in combination with VEN in individuals under the age of 65 [217]. The combination of HMA (hypomethylating agents) and VEN has demonstrated considerable potential. In patients who are deemed ineligible for intensive chemotherapy, the second phase III VIALE-A trial conducted a comparison between AZA and AZA/VEN [218]. The extended OS of AZA/VEN exhibits a greater degree of extension compared to that of AZA [219]. NPM1c AML has a good prognosis without FLT3-ITD mutations [220]. FLT3 is still an appropriate target for the treatment of AML, although the therapeutic efficacy of single medicines has been constrained by transient responses and resistance [221]. Due to their widespread availability and poor prognosis, FLT3 inhibitor therapy for people with FLT3-ITD mutations has gained favor over the past ten years [222]. Based on their capacity to inhibit FLT3 with either an ITD or TKD mutation (type 1 inhibitors) or just an ITD mutation (type 2 inhibitors), FLT3 inhibitors (FLT3i) are classified as either type 1 or type 2 inhibitors. The type 1 inhibitors sorafenib, midostaurin, lestaurtinib, crenolanib, and gilteritinib, and the type 2 inhibitors sorafenib, ponatinib, and quizartinib are components of first-generation FLT3i [219,222]. According to preliminary studies, people who use these drugs may eventually develop an FLT3 TKD mutation as a kind of resistance or an escape mechanism [223]. Treatment options for FLT3-mutated AML include hypomethylating agents (HMAs; AZA, decitabine), low-intensity treatment, or aggressive chemotherapy [224]. The FLT3 mutation, whether it is an internal tandem duplication (ITD) or tyrosine kinase domain (TKD) mutation, was addressed in the initial phase Ib study of HMA combined with VEN, as well as in the VIALE-A trial [218]. In these studies, the combination of azacitidine and VEN demonstrated greater efficacy compared to AZA alone [225]. Additionally, the efficacy of AZA in combination with sorafenib was investigated in a phase II study involving patients with AML harboring FLT3 mutations [226]. The combination exhibited a favorable tolerance profile. Ivosidenib (AG-120) [227], a specific inhibitor for IDH1-mutation, and enasidenib (AG-221), a specific inhibitor for IDH2-mutation [228,229], were recently approved by the FDA for the treatment of AML patients with their respective mutations [230]. The ongoing Phase III AGILE trial is currently investigating the efficacy and safety of two treatment regimens for newly diagnosed acute myeloid leukemia (AML) patients with IDH2 mutations. These regimens involve the use of ivosidenib with or without azacitidine (AZA), as well as enasidenib with or without AZA. The trial aims to evaluate the potential benefits and potential side effects of these treatment approaches in this specific patient population [231]. However, as most AML patients are over 65, they may be at greater risk for early mortality and induction failure. The majority of younger patients with newly diagnosed AML receive intensive therapy with high-dose araC (HiDAC) and anthracyclines. Greater unfavorable characteristics, comorbidities, and early organ damage were present in older AML patients [14]. Ongoing research and recently approved AML agents are intriguing, especially in combination with VEN, HMAs, and epigenetic therapy in elderly patients [224]. The majority of elderly individuals who have recently been diagnosed with AML typically undergo treatment involving the administration of an HMA or low-intensity chemotherapy, with VEN providing the foundational therapeutic approach [219]. In this phase III trial, older patients with newly diagnosed AML were treated with decitabine, SC, and low-dose cytarabine. Due to common medication risks, older AML patients have few treatment alternatives [232]. It is possible, however, that TP53-mutated patients have a diminished capacity to undergo apoptosis in response to DNA damage, which diminishes their sensitivity to chemotherapy [233,234]. Previously, HMA monotherapy was used to treat these patients. HMA and VEN together have increased complete (CR) rates, but overall survival (OS) remains poor [218]. Unfortunately, there are not many choices for treating older patients with TP53-mutated AML with currently approved medications [235]. Resolving the ongoing clinical challenges experienced by elderly AML patients is challenging due to the increased prevalence of medical comorbidities and disease symptoms among the elderly [219]. As depicted in Figure 5, our focus will be on both the FDA-approved treatment methods currently in use and the forthcoming therapeutic interventions that will be investigated primarily through clinical trials.

## 8. Conclusions and Perspectives

Due to its high toxicity, patients and doctors frequently choose palliative care over standard chemotherapy reinduction [236]. Multiple genes and signaling pathways are involved in early and late AML due to its complexity [237]. Utilizing cell lines for decades has enabled extensive, reproducible, and cost-effective AML biology research. Abnormal cell lines facilitate research on diseases, AML subtypes, and gene mutations [238]. Given the complexity and diversity of AML genes, system biology is particularly beneficial in network pharmacology in this instance because it can analyze biological networks and select signal nodes for the development of drugs with multiple targets. Pharmacology networks regulate multichannel signal pathways. This improves drug efficacy and decreases adverse effects, which increases clinical trial success and decreases drug development costs [239,240]. It improves pharmaceutical understanding and drug discovery [187,188]. A unique approach to understanding herbal medicine efficacy regulation, targets, and diseases is developing alongside network pharmacology in AML [241]. Network pharmacology research may uncover drug–disease interactions, such as herbal healing with several active ingredients targeting a wide network [242]. Thus, herbal medicine generated from plants capable of treating a variety of diseases produces efficacious natural components for commercial use [205]. Traditional research is limited in its scope. Since the majority of tertiary academic cancer patients were from the West, conventional treatment was less beneficial for Eastern cancer patients. Expression profiling is one of the best molecular methods for prognosis and prediction, particularly when combined with other data, despite its limitations [243]. It is unknown if patients who obtain remission have deeper remissions and can tolerate additional consolidation therapies, despite the fact that the rate of remission induction for FDA-approved drugs is extraordinarily high [244,245]. In recent years, there has been an intensive exploration of novel treatments for AML with the objective of enhancing patient survival. Novel therapeutic approaches encompass a range of interventions, such as targeted gene therapies, cytokine therapies, and immunotherapies. Immunotherapy, a therapeutic approach that harnesses the inherent capabilities of the patient’s immune system to target and eradicate leukemic cells, exhibits considerable promise [246,247,248]. Different metabolic reactions are present in AML because of its complexity. The characteristics of the patient and the disease should direct the treatment. Herbal medicines possess significant bioactivity and minimal cytotoxicity because of the variety of compounds they contain. Innovative treatments based on natural products are required for AML and its effects. This review demonstrates that network pharmacology can provide a reliable evaluation of drug candidate function [249]. The utilization of herbal medicines and natural chemicals can then be established by investigating their absorption and bioavailability in animals. A number of compounds with anti-cancer characteristics were found in the extracts after chemical analysis, opening up interesting new study directions. As plants are an important source for the development of novel chemotherapeutic drugs, this study paves the way for the identification of numerous natural chemicals that have the ability to prevent and treat cancer [250]. Using network pharmacology to investigate the pharmacological effects of herbal medicines and natural compounds on AML, the researcher anticipates achieving a major breakthrough toward improved older AML patient therapies.

## Figures and Tables

**Figure 1 ijms-24-12037-f001:**
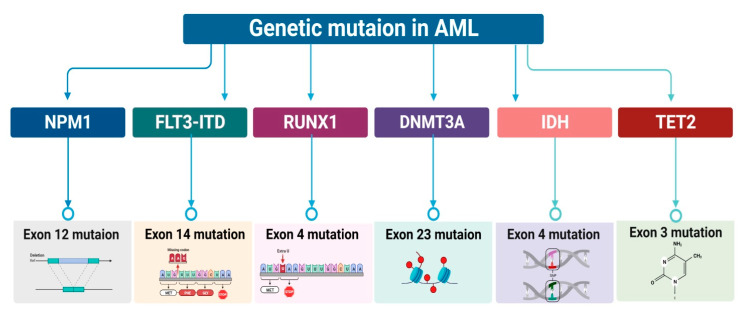
An overview of the key molecules involved in the development of AML. Exon 12 mutations of the NPM1-DIM gene result in changes to the C-terminal amino acids [21], thereby impairing the nucleocytoplasmic transport system. It has been determined that Exon 14 of the FLT3-ITD gene contains in-frame mutations [27]. The RUNX1 frameshift mutation in Exon 4 (c.497_498insGA) has been identified [32]. The topic of interest is associated with R882-DNMT3A variants. R882 mutation, TET2 exon 3 mutations, and IDH1 exon 4 arginine substitution missense variants are the identified mutations within Exon 23 [43].

**Figure 2 ijms-24-12037-f002:**
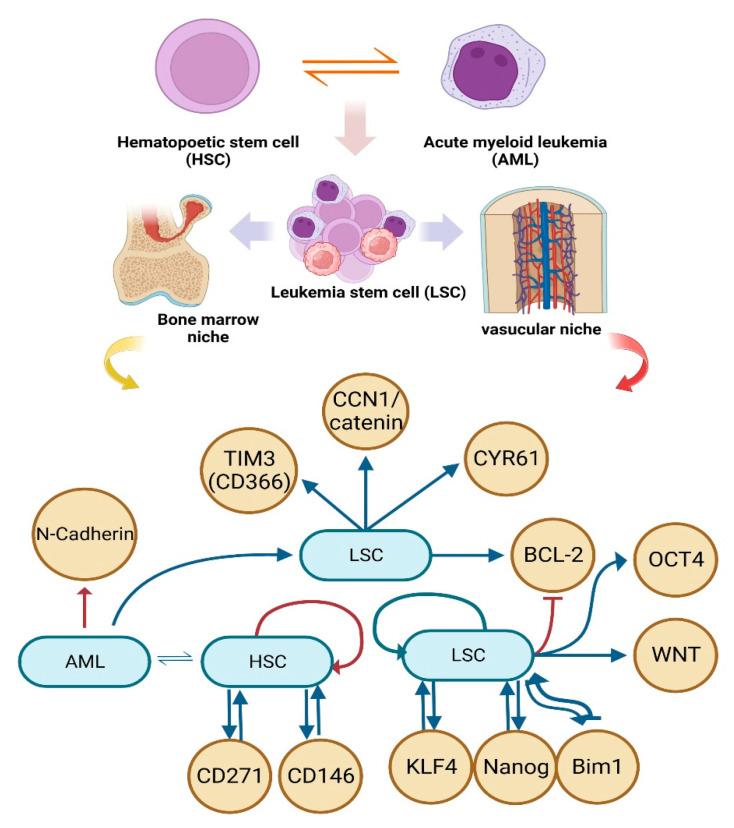
An overview of the molecular crosslinks produced by the interaction of the HSCs and LSCs. AML cells infiltrate the vascular niche of HSCs and directly influence the LSCs that initiate the disease [60,61]. In the population of somatic cells with stem cell-like features (LSCs), extracellular matrix (ECM) components forced tumor-promoting stem cell signaling pathways like KLF4, Bmi-1, Wnt, Nanog, and Oct4, propagating metastasis [90,91,92]. HSCs encompass a range of molecules that play crucial roles in cellular processes. These markers include the T cell marker TIM3 (also known as CD366), which is associated with exhaustion or dysfunction of T cells [63]. Another marker is Cysteine-rich angiogenic inducer 61 (CYR61), β-catenin, CD271, and CD146, proteins involved in promoting angiogenesis, the formation of new blood vessels in HSCs [86,87,88].

**Figure 3 ijms-24-12037-f003:**
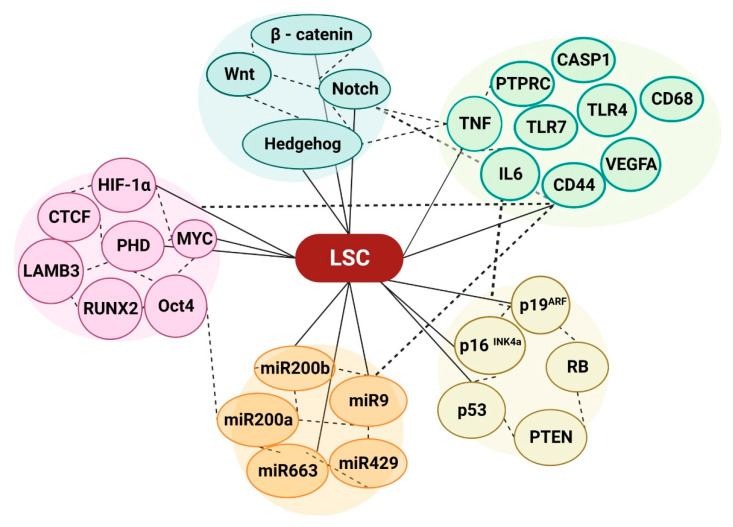
An overview of the LSCs’ microenvironment in relation to epigenetic modification. The Wnt/β-catenin, Hedgehog, and Notch signaling pathways are responsible for the regulation of cancer stem cells within a tumor. These pathways play a crucial role in the ability of cancer stem cells to undergo self-renewal and differentiate into various cell types [,[108],[109],[110],[111]]. The regulation of cell cycle entry and exit is governed by several key factors, namely p16INK4a, p19ARF, RB, and p53 [113]. These crucial components play a pivotal role in maintaining the delicate balance of cell division and preventing the onset of cancer. The genes that have been identified as the hub genes associated with the recurrence of AML are TNF, IL6, TLR4, VEGFA, PTPRC, TLR7, TLR1, CD44, CASP1, and CD68 [126]. The modulation of various biological entities, such as the transcription factor HIF-1α, the DNA-binding protein CTCF motif, the proto-oncogene MYC, LAMB3, RUNX2, and Oct4, is influenced by the presence of CpG-methylated recognition site sequences [129,130]. The miR-200, miR-9, and miR-200b-200a-429 molecules hold significant significance in the intricate mechanism of DNA methylation regulation within cancer stem cells [131,132,133,134,135].

**Figure 4 ijms-24-12037-f004:**
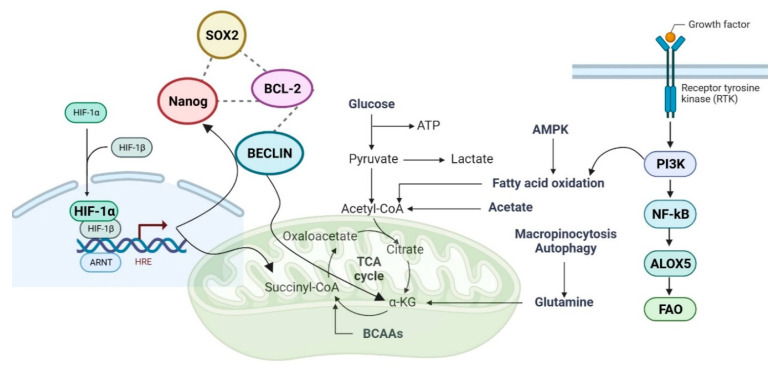
An overview of the LSCs mechanism that employs fat as fuel. The LSCs are responsible for maintaining the process of lipolysis in adipocytes. This metabolic pathway leads to the breakdown of stored lipids, resulting in the production of free fatty acids. These free fatty acids play a crucial role in enhancing the process of β-oxidation within the cells. Alpha-ketoglutarate is important in providing the building blocks for the synthesis of collagen and amino acids, such as glutamine and glutamate, which are required for these metabolic activities. Glutamine is a necessary ingredient for the survival of cancer stem cells. In hypoxia, a condition characterized by insufficient oxygen supply, it is observed that Nanog exhibits a direct binding affinity toward the enhancer region of the BNIP3L promoter. This interaction leads to the release of BECLIN from its association with Bcl-2. Consequently, this dissociation facilitates an upregulation in the expression of BECLIN.

**Figure 5 ijms-24-12037-f005:**
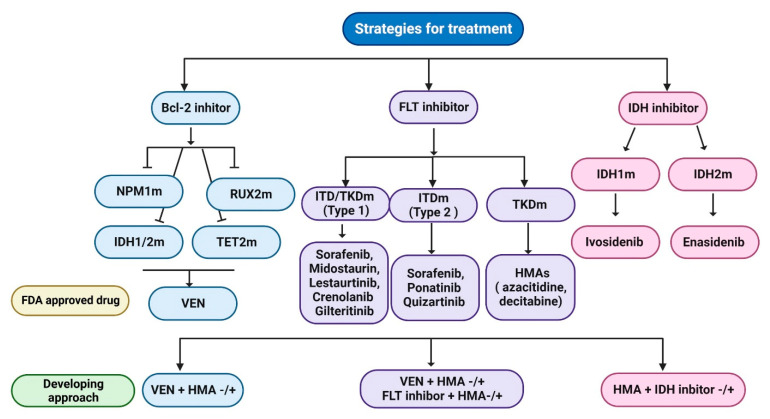
Using presently available, approved drugs and clinical trials as a model. NPM1-, IDH1/2-, TET2-, and RUNX1-m are all optimized by VEN. AZA and AZA/VEN received comparisons in patients receiving ineligible intense chemotherapy in the phase III VIALE-A study. Depending on whether they have an ITD or TKD mutation, FLT3 inhibitors (FLT3i) are classified as type 1 or type 2. The FLT3 mutation—ITD or TKD—was addressed in the phase Ib research with HMA and VEN. The inhibitors of IDH1 and IDH2 are ivosidenib (AG-120) and enasidenib (AG-221). In clinical trials, these regimens combine enasidenib and ivosidenib with or without AZA.

**Table 1 ijms-24-12037-t001:** The application of herbal medicine with apoptotic properties.

Classification	Compound	Effect (Cell)	Reference
Flavonoid	*Leonurus japonicus* Houttuyn (LJH)	miR-19a-3p/PTEN (THP-1, U937)	[191]
	*Daemonorops draco* Blume (DD)	miR-216b/c-Jun (THP-1, U937)	[192]
	Casticin	miR-338-3/RUNX1 PI3K-AKT (HL-60, HS-5)	[193]
Arsenic-Containing	Qinghuang (QHP)	AKT1, MAPK1, MAPK3, PIK3CG CASP3, CASP9, TNF, TGFB1, MAPK8, and TP53 (KG1-a, HL-60) NF-kB, p53, ROS	[194]
Flavonoid	Parthenolide (PTL)	primary AML cells, bcCML cells, normal bone marrow (BM), and umbilical cord blood (CB) cells from the National Disease Research Interchange (NDRI)	[195]
	Quercetin	caspase-8, caspase-9, and caspase-3, cleaved PARP, and depolarized mitochondria via ERK-mediated apoptosis pathway (THP-1, MV4-11, U937, HL-60 cells)	[196]
	1-Cinnamoyltrichilinin	p38 pathway	[197]
Alkaloid	Curine	Cell cycle arrest (HL-60 cells)	[198]
Carotenoid	Indicaxanthin	NOX-4, NF-κB, redox balance, Ca homeostasis, mitochondrial damage (THP-1 cells)	[200]
	Fucoxanthin	cleaved caspases-3, -7, PARP, Bcl-xL, ROS (HL-60 cells)	[201]
	Isolancifolide	cytochrome c, Smac/DIABLO, and cleaved caspases-8, -3, and PARP (HL-60 cells)	[202]
Phenolics	Curcumin	ERK/JNK/Jun cascade (THP-1 cells)	[204]

## Data Availability

Not applicable.

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
