# Peer review of "The Therapeutic Potential of a Strategy to Prevent Acute Myeloid Leukemia Stem Cell Reprogramming in Older Patients"

_ijms, 2023, doi:10.3390/ijms241512037_

Round 1

Reviewer 1 Report

In the manuscript entitled "The therapeutic potential of herbal remedies to prevent leukemia stem cell reprogramming in acute myeloid leukemia", the authors have discussed an overview of the characterization of genetic mutation in acute myeloid leukemia, herbal treatment strategies for acute myeloid leukemia and clinical research in acute myeloid leukemia. This manuscript is written well and is of great importance for scientists working in the area of drug discovery of anticancer agents, particularly acute myeloid leukemia. The manuscript needs some minor changes before being accepted for publication. 

Specific comments:

1. Figures not cited in the text. 

2. List herbal products discussed in the section "Herbal Treatment Strategies for AML Employing System Biology" in Table. 

3. In the section "Clinical research and AML treatment", include details of clinical trials (ongoing) of herbal-based products for the treatment of acute myeloid leukemia.

Author Response

In the manuscript entitled "The therapeutic potential of herbal remedies to prevent leukemia stem cell reprogramming in acute myeloid leukemia", the authors have discussed an overview of the characterization of genetic mutation in acute myeloid leukemia, herbal treatment strategies for acute myeloid leukemia and clinical research in acute myeloid leukemia. This manuscript is written well and is of great importance for scientists working in the area of drug discovery of anticancer agents, particularly acute myeloid leukemia. The manuscript needs some minor changes before being accepted for publication. 

 >> (Response) First of all, I would like to express my sincere gratitude for the time and effort the reviewer had put into reviewing our manuscript. I have incorporated changes based on the reviewer comments provided in the manuscript which revised parts are highlighted by blue color in the entire revised manuscript.

Specific comments:

  1. Figures not cited in the text. 

>> (Response) I am grateful the reviewer nice comments. I modified.

  1. List herbal products discussed in the section "Herbal Treatment Strategies for AML Employing System Biology" in Table. 

>> (Response) I am grateful the reviewer nice comments. I added in table 1.

  1. In the section "Clinical research and AML treatment", include details of clinical trials (ongoing) of herbal-based products for the treatment of acute myeloid leukemia.

>> (Response) I am grateful the reviewer nice comments. I added the mentioned meaningful suggestions in section "Clinical research and AML treatment" and in Figure 5.

Reviewer 2 Report

The current review article illustrates the potential of traditional herbal medicines as anticancer agents against AML and their novel mechanistic actions. The current draft has briefly encompassed key genetic mutations and epigenetic modifications in AML, followed by an overview of the anti-cancer effects of herbal medicine and signaling pathways associated with this disease. In concert, studies included by the author in this work hold the potential of contributing significantly toward the understanding of AML phenotype and treatment strategies. For these reasons, this work is suitable for publication in IJMS. It meets the impact criteria of this journal because the work will be of significant interest to a broad readership in the chemical biology and medicinal chemistry community. However, a few major issues should be addressed beforehand, as listed below.

While the author has categorically presented the key literature on these topics, the writing style should be significantly improved in the first two sections (namely, Characterization of Genetic mutation in AML and Epigenetic modification of the LSC Microenvironment). The author should classify these sections into sub-sections for a better understanding of the readers. A few sentences are redundant and hamper the coherence of the results presented (e.g., line 125-line 131). Hence, overall, these sections should be carefully overviewed by the author and modified as needed.

The author has only included flavonoids as herbal medicines exhibiting anticancer effects against AML in the current article. However, a large variety of natural products including carotenoids and alkaloids are known to exhibit cytotoxic effects against AML cells. Phenolic compounds such as curcumin, found in turmeric, can induce apoptosis of human monocytic leukemia THP-1 cells via the activation of JNK/ERK Pathways. The author should include those.

The clinical progress currently available for AML patients has been mentioned. As the article revolves around the use of herbal medicines as treatment strategies, the potential side effects of these agents and further improvement strategies should also be briefly discussed. The author should also mention the FDA-approved drugs for AML.

Immunotherapy posits as a future treatment approach for AML. The reviewer is curious if herbal medicines have been investigated for exhibiting immunotherapeutic effects in AML cells. The author can briefly discuss that in the current article.

As mentioned in the review report, the quality of English and the writing style do not fit completely the standard of the article and should be improved. 

Author Response

The current review article illustrates the potential of traditional herbal medicines as anticancer agents against AML and their novel mechanistic actions. The current draft has briefly encompassed key genetic mutations and epigenetic modifications in AML, followed by an overview of the anti-cancer effects of herbal medicine and signaling pathways associated with this disease. In concert, studies included by the author in this work hold the potential of contributing significantly toward the understanding of AML phenotype and treatment strategies. For these reasons, this work is suitable for publication in IJMS. It meets the impact criteria of this journal because the work will be of significant interest to a broad readership in the chemical biology and medicinal chemistry community. However, a few major issues should be addressed beforehand, as listed below.

 >> (Response) First of all, I would like to express my sincere gratitude for the time and effort the reviewer had put into reviewing our manuscript. I have incorporated changes based on the reviewer comments provided in the manuscript which revised parts are highlighted by blue color in the entire revised manuscript.

While the author has categorically presented the key literature on these topics, the writing style should be significantly improved in the first two sections (namely, Characterization of Genetic mutation in AML and Epigenetic modification of the LSC Microenvironment). The author should classify these sections into sub-sections for a better understanding of the readers. A few sentences are redundant and hamper the coherence of the results presented (e.g., line 125-line 131). Hence, overall, these sections should be carefully overviewed by the author and modified as needed.

>> (Response) I am thankful to the reviewer for the meaningful suggestions. I massively checked the mentioned section English language and reviewed to improve the quality of my manuscript. 

The author has only included flavonoids as herbal medicines exhibiting anticancer effects against AML in the current article. However, a large variety of natural products including carotenoids and alkaloids are known to exhibit cytotoxic effects against AML cells. Phenolic compounds such as curcumin, found in turmeric, can induce apoptosis of human monocytic leukemia THP-1 cells via the activation of JNK/ERK Pathways. The author should include those.

>> (Response) I am grateful the reviewer nice comments. I modified.

The clinical progress currently available for AML patients has been mentioned. As the article revolves around the use of herbal medicines as treatment strategies, the potential side effects of these agents and further improvement strategies should also be briefly discussed. The author should also mention the FDA-approved drugs for AML.

 >> (Response) I am grateful the reviewer nice comments. I added the mentioned meaningful suggestions in Figure 5.

Immunotherapy posits as a future treatment approach for AML. The reviewer is curious if herbal medicines have been investigated for exhibiting immunotherapeutic effects in AML cells. The author can briefly discuss that in the current article.

>> (Response) I am grateful the reviewer nice comments. I added in discussion.

Reviewer 3 Report

The manuscript entitled " is all about Acute myeloid leukemia. The importance and details of herbal natural products are very brief and generic. In my view, the manuscript title will mislead the readers. 

The natural products included in the manuscript like Casticin, Quercetin, etc are well-documented for anticancer properties. 

The manuscript is not proving substantial information or analysis about herbal natural products for AML to the scientific community.

The manuscript is NOT SUITABLE for a journal like IJMS in its current form.

Author Response

The manuscript entitled " is all about Acute myeloid leukemia. The importance and details of herbal natural products are very brief and generic. In my view, the manuscript title will mislead the readers. 

 >> (Response) First of all, I would like to express my sincere gratitude for the time and effort the reviewer had put into reviewing our manuscript. I have incorporated changes based on the reviewer comments provided in the manuscript which revised parts are highlighted by blue color in the entire revised manuscript. I modified title.

The natural products included in the manuscript like Casticin, Quercetin, etc are well-documented for anticancer properties. 

>> (Response) I am grateful the reviewer nice comments.

The manuscript is not proving substantial information or analysis about herbal natural products for AML to the scientific community.

>> (Response) I am grateful the reviewer nice comments. I added clear information or analysis about herbal medicine for AML in table 1.

The manuscript is NOT SUITABLE for a journal like IJMS in its current form.

>> (Response) I appreciate the reviewer's meaningful suggestions. I massively checked and reviewed to improve the quality of my manuscript.

Round 2

Reviewer 2 Report

The author has incorporated the suggested changes. The manuscript is ready for publication.

The quality is now better than the previous version.

Reviewer 3 Report

The manuscript is now revised and restructured extensively.